# The Pharmacological Efficacy of Baicalin in Inflammatory Diseases

**DOI:** 10.3390/ijms24119317

**Published:** 2023-05-26

**Authors:** Yongqiang Wen, Yazhou Wang, Chenxu Zhao, Baoyu Zhao, Jianguo Wang

**Affiliations:** College of Veterinary Medicine, Northwest A&F University, Xianyang 712100, China; wyq08137439@nwafu.edu.cn (Y.W.); wangyazhou@nwafu.edu.cn (Y.W.); cxzhao@nwafu.edu.cn (C.Z.); zhaobaoyu12005@163.com (B.Z.)

**Keywords:** baicalin, bioavailability, drug interaction, anti-inflammatory activity, pharmacokinetics

## Abstract

Baicalin is one of the most abundant flavonoids found in the dried roots of *Scutellaria baicalensis* Georgi (SBG) belonging to the genus *Scutellaria*. While baicalin is demonstrated to have anti-inflammatory, antiviral, antitumor, antibacterial, anticonvulsant, antioxidant, hepatoprotective, and neuroprotective effects, its low hydrophilicity and lipophilicity limit the bioavailability and pharmacological functions. Therefore, an in-depth study of baicalin’s bioavailability and pharmacokinetics contributes to laying the theoretical foundation for applied research in disease treatment. In this view, the physicochemical properties and anti-inflammatory activity of baicalin are summarized in terms of bioavailability, drug interaction, and inflammatory conditions.

## 1. Introduction

*Scutellaria baicalensis* Georgi (SBG), belonging to the genus *Scutellaria*, grows mainly in Asia, including China, Mongolia, Japan, Korea, and Siberia [1]. In China, SBG is regarded as an authentic traditional medicinal herb that grows widely in desert areas and sunny grassy slopes at altitudes of 60–2000 m [2]. It is mainly found in northern provinces of China such as Hebei, Shandong, Shanxi, and Inner Mongolia [3]. Based on SBG’s growing season and cycle characteristics, Xu et al. (2020) determined that the quality of SBG harvested in autumn and cultivated for three years was the best [2]. The primary active constituents of SBG encompass baicalin, baicalein, wogonin, Han baicalin, and Han baicalein [4]. In the field of clinical practice, *Scutellaria baicalensis* Georgi is frequently employed to treat inflammatory disorders, influenza, diarrhea, jaundice, headache, and abdominal pain [5]. The broad-spectrum pharmacological effect of SBG in diminishing various kinds of diseases is mainly through regulating host immunity [6]. In addition, SBG was reported to have effects such as enhancing immunity, anti-aging, protecting the liver, and anti-osteoporosis [7,8].

Flavonoids are a class of compounds that are widely distributed in various vegetables and fruits [9]. The consumption of flavonoid-rich fruits and vegetables could reduce the risk of inflammatory diseases [9]. In vivo and in vitro analyses have confirmed that certain flavonoids play a therapeutic effect on diseases. For example, baicalin (7-glucuronic acid, 5,6-dihydroxy-flavone, C_21_H_18_O_11_) has been widely used in recent years for the development of pharmaceutical formulations and the treatment of certain diseases [10]. Baicalin, also known as begalin, is formed by combining the C7 hydroxyl group of baicalein with glucuronic acid [10] (Figure 1). Baicalin is a light-yellow powder under normal conditions; it is bitter in taste, insoluble in alcohols, and soluble in chloroform, nitrobenzene, dimethyl sulfoxide, etc. [11,12,13]. For the function of baicalin, studies have demonstrated that baicalin, containing most of the pharmacological function of SBG in modulating host immunity, plays a therapeutic role in neuroinflammation, enteritis, pneumonia, secondary inflammation, and other diseases in the clinic [14,15,16]. Inflammation is an immune response to invasiveness, which aims to clear away invasive pathogens and initiates tissue repair [17]. Although inflammation has this protective function, inappropriate inflammation can trigger damage to the body and induce the development of diseases [18]. Baicalin has been demonstrated to have anti-inflammatory and immunomodulatory functions, most of which are regulated by inhibiting the activation of nuclear factor kappa-light-chain-enhancer of activated B cells (NF-κB) signaling pathway and nucleotide-binding oligomerization domain-like receptor pyrin domain protein 3 (NLRP3) inflammasome as well as suppressing pro-inflammatory factor expression, such as interleukin (IL)-1β, IL-6, IL-8, tumor necrosis factor α (TNF-α), cyclooxygenase 2 (COX-2), inducible nitric oxide synthase (iNOS), etc. [1,17].

In summary, the pharmacological effects of baicalin are closely related to inhibiting inflammatory reactions. In this paper, the basic physicochemical properties of baicalin and its molecular and immune regulatory mechanisms in the prevention and treatment of inflammatory diseases are reviewed.

## 2. Bioavailability of Baicalin

In pharmacological studies, bioavailability refers to the degree of drug absorbed into the systemic circulation, reflecting the percentage of drug absorbed by the gastrointestinal tract to the oral amount [19]. Generally, the permeability coefficient of drugs with 1% absorption is about 1.0 × 10^−6^ cm/s. The permeability coefficient of drugs absorbed between 1% and 100% ranges from 1.0 × 10^−6^ to 0.1 cm/s, and the permeability coefficient of drugs and peptides absorbed less than 1% is below 1.0 × 10^−7^ cm/s [20].

Its low water solubility (67.03 ± 1.60 μg/mL) and permeability (0.037 × 10^−6^ cm/m) determine that baicalin cannot be transported by passive diffusion into the host cell lipid bilayer, which results in poor absorption and the low bioavailability of baicalin [21]. Contrarily, baicalein, a glycoside form of baicalin with good permeability and lipophilicity, can be well absorbed by the gastrointestinal tract [22]. Studies have shown that there is an interconversion between baicalin and baicalein during the absorption process of baicalin [22,23,24]. In particular, after drug administration in animals, baicalin could be hydrolyzed to baicalein by β-glucuronidase derived from intestinal bacteria, and baicalein could be recovered to baicalin by uridine 5′-diphosphate (UDP) glucuronide transferase (UGT) circulating in vivo [25]. This mutual conversion maximizes the full pharmacodynamic functions of baicalin [25]. Due to the better absorption of baicalein compared to baicalin, the conversion of baicalin to baicalein is a key step for the absorption of baicalin in the intestine. Furthermore, the intestinal microbiota was reported to be associated with the conversion of baicalin to baicalein [26]. For instance, the intestinal absorption of baicalin in germ-free rats was significantly reduced compared to conventional rats [26], suggesting that the presence of intestinal flora was conducive to the body’s absorption and utilization of baicalin. In other words, only a small proportion of baicalin is absorbed by the physical body as a raw component, and most of it is hydrolyzed into baicalein by bacteria and absorbed by the physical body. Human serum albumin (HSA) is the main transport medium through which flavonoids (for example, baicalin, catechin, quercetin, etc.) are absorbed and utilized by the physical body [27]. The rate of transport and volume distribution of baicalin in the host depends on the binding degree of baicalin to HSA [28]. Especially, it was confirmed that, based on the area under the time–concentration curve, the relative absorption rate of baicalin was approximately 65% [29]. Further pharmacokinetic analysis showed that the peak concentration of baicalin in rat serum was lower than that of baicalein [30]. To improve the bioavailability of baicalin, investigators focused on the development of new formulations of baicalin, such as solid nanocrystal nano-emulsions, solid–liquid nanoparticles, liposome formulations, phospholipid complex hydrogels, etc. [31,32,33]. The application development of these new technologies and products aims to improve the solubility and absorption of baicalin.

## 3. Toxicity of Baicalin

Clearing the safe dose range and action time of drugs is crucial for drug properties, as these directly threaten animals’ safety. The determination of baicalin’s safe dosage range and its duration of action holds paramount significance across diverse animal and cellular models [34]. To investigate whether baicalin was involved in the regulation of hepatic insulin resistance and gluconeogenic activity, an ex in vivo study showed that intraperitoneal injection of baicalin (50 mg/kg) not only resulted in body weight loss and insulin resistance (Homeostatic model assessment for insulin resistance, HOMA-IR) in obese mice (C57BL/6J) but also reduced glucose intolerance and hyperglycemia. In this study, they also showed that baicalin was not toxic to mice’s liver [35]. In line, baicalin was found to effectively alleviate the development of obesity by inhibiting the expression of p-p38 mitogen-activated protein kinase (MAPK), phosphorylation cyclic adenosine 3′,5′-monophosphate (cAMP) response binding protein (p-CREB), forkhead transcription factor forkhead box O1A (Foxo1), peroxisome proliferator-activated receptor γ coactivators 1α (PGC-1α), phosphoenolpyruvate carboxykinase (PEPCK), and glucose-6-phosphatase (G6Pase) in the liver of obese mice and hepatocytes [35]. Another study performed by Shi et al. (2020) demonstrated that liver toxicity induced by acetaminophen (APAP) in mice was effectively reduced at 6 h, 12 h, and 18 h after baicalin administration, and the necrotic area of liver cells was significantly reduced in mice administrated with 40 mg/kg baicalin [36]. These results indicate that baicalin at a safe dose not only has no cytotoxicity in mice but can also reduce lipotoxicity. In line with this, baicalin was reported to effectively inhibit the occurrence of lipopolysaccharide (LPS)-induced hepatitis in chickens treated with 50, 100, and 200 mg/kg baicalin [37]. An oxidative stress study in non-alcoholic fatty liver disease (NAFLD), revealed that baicalin at concentrations ranging from 0.01 nM to 100 μM did not exhibit any cytotoxic effects on HepG2 cells at 24 h and 48 h, as determined by the cholecystokinin (CCK)-8 assay [38].

Baicalin has a significant inhibitory effect on endoplasmic reticulum (ER) stress. For instance, baicalin (12.5 μM and 25 μM) was found to effectively inhibit the expression of ER stress marker phosphorylated inositol-requiring enzyme 1α (p-IRE1α) and reverse palmitic acid (PA)-induced apoptosis and reactive oxygen species (ROS) production [38]. The main mechanism might be because baicalin reduced PA-induced cytotoxicity by inhibiting ER stress and the activation of thioredoxin-interacting protein (Txnip)/NLRP12 inflammasome [39]. These studies provide a new theoretical basis for the clinical application of baicalin and further improve the cytotoxicity study of baicalin. From a toxicological point of view, baicalin is more toxic than baicalein [39]. Interestingly, intestinal microorganisms might have a protective effect on hepatotoxicity caused by baicalin [39]. To support this, incubation of baicalin with fecal enzymes diminished the cytotoxicity to hepatocellular carcinoma (HepG2) cells in a dose-dependent manner, suggesting that the conversion from baicalin to baicalein by human fecal enzymes protects against baicalin-induced cytotoxicity to HepG2 cells [40,41]. These studies pave a new theoretical basis for the clinical application of baicalin and further improve the cytotoxicity studies of baicalin.

## 4. Interaction between Baicalin and Other Drugs

Multi-drug combination therapy is now a therapeutic schedule for various diseases. Therefore, a thorough understanding of baicalin and other drug interactions is required before taking medication.

Baicalin is widely used in the prevention and treatment of clinical diseases. Drug interactions can occur when baicalin is taken together with other drugs, and this situation warrants further attention. It was found that the intestinal flora plays a key role in the absorption and utilization of drugs [42]. Baicalin can be structurally converted to baicalein by intestinal bacteria and related enzyme action, thus affecting the efficacy of baicalin [43]. Therefore, when combined administration of baicalin with antibiotics is adopted, the effect of antibiotics on the pharmacokinetic properties of baicalin by inhibiting intestinal bacteria should be considered [44]. Drug conversion between baicalin and baicalein is associated with various transporters and metabolic regulatory enzymes, which may be affected by co-administration [25]. The pharmacokinetic effects of baicalin combined administration show a common feature: baicalin can significantly alter drug pharmacokinetics with the same cytochrome P450 (CYP) enzyme activity or high protein binding [45]. Concretely, the effects of baicalin on phenacetin [46], theophylline [47], midazolam [48], dexamethasone [49,50], nifedipine [51,52], and rosuvastatin [53] involve the metabolism through CYP 1A2, CYP2E1, CYP3A, and CYP2D [47]. Baicalin was also found to alter the pharmacokinetics of nifedipine, another CYP3A substrate, in a different way [54]. The binding of drugs such as theophylline, nifedipine, and promethazine to HSA is decreased in the presence of baicalin [55]. The main reason is that HSA is a competition site, resulting in competition with each other [56]. When co-administered, the binding rate of baicalin to HSA may be altered, and baicalin may inhibit CYP enzyme activity; therefore, the complexity of co-administration must be further explored to reduce adverse drug reactions during clinical co-administration [51,57,58].

## 5. Therapeutic Effects of Baicalin

### 5.1. Effect of Baicalin on Hepatitis

TNF-α is an important cytokine involved in metabolic diseases such as obesity, insulin resistance, hyperlipidemia, and NAFLD [59]. The production of TNF-α in the process of liver injury causes the release of IL-1β and IL-6, which mediates inflammatory response [60]. Hepatotoxicity could activate the toll-like receptor 4 (TLR4) through excessive lipid accumulation in hepatocytes [61]. Upon activation of TLR4, a large amount of MyD88 aggregates NF-κB and MAPK [62]. The aggregations of NF-κB and MAPK promote the expression of TNF-α, IL-1β, IL-6, and other pro-inflammatory factors and the infiltration of pro-inflammatory cells, which further aggravates liver injury [63]. For instance, studies showed that baicalin could reduce the expressions levels of TNF-α and NF-κB in liver tissue and the production of TNF-α, IL-1β, and IL-6 in plasma, thus reducing the degree of liver inflammation [64,65]. In line with this, baicalin reduced systemic inflammation by blocking the NLR pyrin domain containing 3-Gasdermin D signal transduction and by inhibiting TLR4 signal cascade in mice, reducing the release of pro-inflammatory factors (TNF-α, IL-8, and IL-6) [36,37,66]. In addition, studies found that baicalin can initiate the repair of liver injury caused by acetaminophen (APAP) [67]. The main mechanism is to promote liver regeneration after APAP-induced acute liver injury in mice by inducing the accumulation of nuclear factor erythroid2-related factor 2 (NRF2) in the cytoplasm and the activation of NLRP3 inflammasome, which in turn leads to the increase in IL-18 expression and the proliferation of hepatocytes, thus achieving liver-regeneration function [36,67].

### 5.2. Role of Baicalin in Rheumatoid Arthritis

Rheumatoid arthritis (RA) is a chronic inflammatory disease in the synovium, which can lead to cartilage and bone damage and disability [68]. Factors involved in the development of diseases include genetic factors, infections, and immune dysfunction [69,70]. Currently, RA is managed clinically with non-steroidal anti-inflammatory drugs (NSAIDs), anti-rheumatic drugs (DMARDs), immunosuppressive drugs, etc. [71]. During the onset and spread of RA, immune T and B lymphocytes activate the effector cells and then release pro-inflammatory mediators such as IL-1, IL-6, IL-17, and TNF-α, which are primarily responsible for synovial joint inflammation and bone erosion [71]. IL-17 is an important cytokine produced by T helper 17 cells (Th17) [72]. As a potent inflammatory cytokine, it could induce the production of a variety of pro-inflammatory factors such as IL-6, TNF-α, and IL-1β, all of which can lead to an inflammatory response in tissues and cells [73] and a significant increase in IL-17 in both the serum and joint fluid of RA patients relative to osteoarthritis [74]. The production of a range of chemokines induced by IL-17 led to the recruitment of T cells, B cells, monocytes, and neutrophils in diseased joints [75]. Matrix metalloproteinases (MMPS), nitric oxide (NO), and nuclear factor-κB (RANK)/RANK ligand (RANKL) receptor activators can be upregulated by IL-17 in both cartilage and osteoblasts, leading to damage in bone and articular cartilage and promoting the development of RA [76,77]. Therefore, inhibition of IL-17 expression might be an important target to improve RA (Figure 2).

Inactivated *Mycobacterium tuberculosis* adjuvant-induced arthritis in mice showed that intraperitoneal administration of 100 mg/kg baicalin significantly inhibited the expansion of the spleen Th17 cell population (40%) and attenuated arthritic symptoms such as paw and ankle swelling [79]. A molecular mechanism study showed that baicalin inhibited the expression of the retinoid-related orphan nuclear receptor γt (RORγt) gene, a key transcription factor for Th17 cell differentiation [80]. Furthermore, baicalin treatment in IL-17-contaminated synovial cells for 24 h significantly inhibited lymphocyte adhesion to synovial cells, blocked IL-17-induced inflammatory cascade, and reduced the expression of intercellular cell adhesion molecule-1 (ICAM-1), vascular cell adhesion molecule (VCAM-1), IL-6, and TNF-α [79]. Additionally, Tong et al. (2018) found that baicalin reduced TLR2 and MyD88 gene expression and the toll-like receptor 2 (TLR2), myeloid differentiation factor 88 (MyD88), and NF-κB-p65 protein expression in RA synovial fibroblasts (RA-FLS) [81], suggesting that the mechanism of baicalin’s action might be related to the inhibition of the TLR2/MyD88/NF-κB signaling pathway. The above studies suggest that baicalin has a positive effect in alleviating RA disease, but more detailed mechanisms of action need to be further investigated.

### 5.3. Anti-Inflammatory Role of Baicalin in Obesity and Type 2 Diabetes

Obesity refers to excessive total or local fat in the body, which is a “metabolic syndrome” with indicators such as abnormal blood glucose, blood fat, blood pressure, and insulin resistance (IR) [82]. Excessive nutrient intake and lack of exercise are likely to cause obesity, which is associated with insulin resistance and an increased risk of type 2 diabetes [83]. It was established that excessive white adipose tissue (WAT) is associated with the occurrence of chronic, low-grade, and systemic inflammation [84]. NF-κB, as a major pro-inflammatory nuclear transcription factor, directly increases the expression level of pro-inflammatory cytokines and chemokines [85]. The inactive state of NF-κB is chelated with a complex of inhibitor-κB (IKB) inhibitor protein family members in the cytoplasm [85]. Upon cellular activation, IκB kinase β (IKK-β) phosphorylates IκB, leading to its ubiquitination and proteolytic degradation. This in turn releases and translocates NF-κB to the nucleus and further activates target gene transcription in the nucleus [85]. Thus, inhibition of the NF-κB signaling pathway in adipose tissue reduces the incidence of chronic, low-grade, and systemic inflammation and confers a protective mechanism against the development and progression of insulin resistance and type 2 diabetes [85].

In most cases, obesity develops from regularly eating a high-fat diet (HFD), and excess obesity can lead to fatty liver disease, known as “NAFLD” [86]. Overexpression of adiponectin, leptin, and TNF-α is a major factor increasing the risk of liver fat accumulation, insulin resistance, pancreatic beta cell dysfunction, and fibrosis [87,88]. Baicalin was shown to reduce the degree of fatty liver degeneration and obesity in a dose-dependent manner, which was attributed to baicalin-dependent inhibition of the hepatic calmodulin-dependent protein kinase kinase beta (CaMKKβ)/Adenosine 5‘-monophosphate -activated protein ki-nase (AMPK)/acetyl-CoA carboxylase (ACC) pathway [12]. Therefore, AMPK activators may be an effective target for the treatment of obesity and type 2 diabetes. Diabetes is an important cause of endocrine and metabolic disorders. Long-term hyperglycemia can cause a variety of chronic complications such as diabetic nephropathy and diabetic retinopathy [89,90]. Studies have shown that baicalin can effectively improve diabetic nephropathy, mainly by activating the NRF2-mediated antioxidant signaling pathway and inhibiting the MAPK-mediated inflammatory signaling pathway [89,90]. In support of this, baicalin was reported to significantly reduce the expression of pro-inflammatory factors IL-1β, IL-6, and TNF-α through the MAPK pathway [89,90]. Because of the important clinical significance of baicalin in the prevention and treatment of obesity and diabetes, more and more attention is focused on baicalin in obesity and diabetes.

### 5.4. Role of Baicalin in Respiratory-Related Inflammation

Over the past few years, respiratory diseases have become more prevalent, posing a significant threat to human health and safety. The commonly occurring respiratory diseases include asthma, idiopathic pulmonary fibrosis (IPF), and pulmonary arterial hypertension (PAH) [91]. Asthma, which is recognized as reversible airway obstruction induced by persistent airway inflammation, is caused by airborne pollutants and genetic predisposition factors [92]. IPF or pulmonary fibrosis (PF), characterized by fibroblast proliferation and collagen deposition in lung tissue structure, is a chronic interstitial lung disease caused by pneumonia, excessive smoking, and/or other factors, leading to structural destruction and respiratory failure [93]. Relevant studies reported that a class of small non-coding RNA molecules, such as microRNA-21 (miR-21) [94], miR-29, and miR-155, all involving the progression of IPF, is upregulated by pro-inflammatory cytokines and transforming growth factor β-1 (TGF-β1) [28,91,95]. Baicalin is an effective treatment in patients with respiratory tract diseases [96]. A study in bleomycin (5 U/kg)-induced PF mice showed that intraperitoneal injection of baicalin (120 mg/kg/d) reduced lung fibrosis, collagen deposition, and hydroxyproline levels in lung tissue through reducing TGFβ1-induced extracellular signal-regulated kinaes 1/2 (ERK1/2) signaling compared with the control [96]. To investigate baicalin’s anti-fibrosis effect, a study in mice with knocked-out adenosine A2a receptor (A2aR), which is an inflammatory regulatory receptor, found that, compared with A2aR-positive mice, A2aR-knockout mice had more severe pulmonary fibrosis and higher expression levels of TGF-β1 and phosphorylated ERK1/2 protein [96]. Based on these results, the authors proposed that baicalin might exert its antifibrotic effect by downregulating the elevated levels of TGFβ1 and pERK1/2 and promoting the expression of inflammatory regulatory gene A2aR [96]. The A2aR gene expressed in macrophages, dendritic cells, T cells, B cells, and epithelial cells is considered a novel regulator of inflammation and tissue repair [97]. PAH is characterized by right ventricular hypertrophy and dysfunction due to the deposition of extracellular matrix (ECM) protein collagen fibers on the walls of pulmonary arterioles, which further leads to constriction of pulmonary blood vessels [98]. In a related baicalin study, the results in rats with chronic hypoxia pretreated with baicalin (30 mg/kg) showed inhibition of p38 MAPK and a downregulation of matrix metalloproteinase 9 (MMP-9) expression in pulmonary arterioles, thus alleviating PAH and pulmonary heart disease (right-side cardiac dysfunction) [97]. In addition, baicalin was reported to reduce the expression levels of IL-1β, IL-6, and TNF-α in lung tissue by regulating the p38 MAPK signaling pathway, suggesting an anti-inflammatory role of baicalin [99].

### 5.5. Therapeutical Effects of Baicalin in Inflammatory Bowel Disease

Inflammatory bowel disease (IBD) is a chronic immune disorder characterized by recurrent episodes of abdominal pain, diarrhea, and purulent stool [100]. IBD mainly includes Crohn’s disease (CD) and ulcerative colitis (UC) [101,102,103]. Baicalin plays a protective role in IBD by inhibiting oxidative stress, immune regulation, and anti-inflammatory properties [104]. To support this, baicalin was found to affect inflammatory processes by regulating autophagy and the NF-κB signaling pathway in intestinal epithelial cells, thereby improving paracellular permeability [105]. In line with this, baicalin was reported to attenuate the inflammatory response by modulating the polarization of M1 macrophages towards the M2 phenotype in a dextran sodium sulfate (DSS)-induced ulcerative colitis model [101]. Similarly, it was found that baicalin downregulated inflammatory cytokines’ expression of IL-1β, TNF-α, apoptotic genes Bcl-2, and caspase-9 in the colon of 2,4,6-trinitrobenzesulfonic acid (TNBS)-induced UC rats in a dose-dependent fashion [106]. The treatment of baicalin in inflammation is mainly mediated by suppressing the inhibitor-κB (IKB) kinase complex (IKK)/IKB/NF-κB signaling pathway [106]. Especially, the administration of baicalin (5–20 mg/kg) in TNBS-induced UC rats downregulated the expression of TNF-α and IL-6 and inhibited the TLR4/NF-κB signaling pathway, thereby alleviating UC [107]. Consistently, baicalin (30–120 mg/kg) was reported to have a therapeutic effect in TNBS-induced UC by promoting the activities of antioxidant enzymes such as superoxide dismutase (SOD), catalase, and glutathione peroxidase (GSH-PX) and by reducing the content of malondialdehyde (MDA) [108]. This study further found that baicalin (100 mg/kg) reduced the production of IL-6, IL-1β, and IL-17 in serum and inhibited the activation of SOD, GSH-PX, and MDA in the UC model under high temperature and humidity [108]. These results suggest that the anti-inflammatory effect of baicalin is attributed to inhibiting the activation of the NF-κB and MAPK signaling pathways.

In addition, baicalin (50–150 mg/kg) administration decreased myeloperoxidase (MPO) activity, NO content, and the expression of TNF and IL-6 in the colon of DSS-induced UC rats [109]. Another study showed that baicalin (100 mg/kg) attenuated DSS-induced UC by blocking the TLR-4/NF-κB-p65/IL-6 signaling pathway and inhibiting inflammatory cytokines’ expression of TNF-α, IL-6, and IL-13 [110]. Consistently, baicalin (10 mg/kg) downregulated the expression of macrophage migration inhibitors monocyte chemoattractant protein (MCP)-1 and macrophage inflammatory protein (MIP)-3a in rat colons of a TNBS-induced UC model [111]. The balance of Th17/regulatory T cells (Treg) has been demonstrated to be associated with UC [112]. For instance, baicalin (20–100 mg/kg) regulated the balance of Th17/Treg by inhibiting MDA and ROS production, reducing GSH and SOD levels, and downregulating the expression of Th17-related factors IL-6 and IL-17 in TNBS-induced UC rats [113,114]. In clinical studies of UC patients, baicalin regulated immune balance and relieved the UC-induced inflammation reaction by promoting the proliferation of CD4(+) CD29(+) cells and modulating immunosuppressive pathways [115].

### 5.6. Anti-Inflammatory Effects of Baicalin in Cardiovascular Diseases

Inflammation is an important link with the pathogenesis of cardiovascular diseases such as myocardial fibrosis, atherosclerosis (AS), and myocardial depression, and it plays a crucial role in the further development of disease [116]. AS, the pathological basis of cardiovascular disease, could cause angina pectoris and myocardial infarction, etc. [117]. The main pathological mechanism involved is an endothelial function, lipid deposition, oxidative stress damage, immune inflammation, etc. [118,119]. Baicalin is widely used in lipid-lowering studies because of its effectiveness in reducing serum total cholesterol (TC), triacylglycerol (TG), and low-density lipoprotein cholesterol (LDL-C) levels [120]. In an apolipoprotein E (APOE)^−/−^ mouse model of a high-cholesterol diet, baicalin reduced the expression of TG and LDL-C [121]. Their further analysis showed that baicalin promoted the proliferation and viability of Treg cells and the expression of TGF-β and IL-10, which improved the progression of atherosclerotic lesions through lipid regulation and immunomodulation [122]. In the development of AS-induced inflammation, activation of NF-κB increased the production of inflammatory factors and chemokines [123]. Therefore, baicalin has a beneficial effect on the development of AS by inhibiting the NF-κB signaling pathway [124]. It was also shown that baicalin mediated the Wnt1/dickkopf-related protein 1 (DKK1) signaling pathway to prevent AS [125]. In addition, the Th17/Treg balance is inextricably linked to the development of AS, and IL-17A secreted by Th17 triggers the onset of AS-induced inflammation and Treg, a protective T cell, which effectively attenuates the inflammatory response [126]. Patients with AS commonly have a Th17/Treg imbalance; specifically, Th17 cells increase, and Treg cells decrease [127]. Studies conducted by Jiang et al. (2019) and Yang et al. (2012) showed that baicalin had a protective effect on Th17/Treg homeostasis, which is associated with its anti-AS role [128,129]. In hypertension studies, baicalin reduced hypertension-induced inflammation (significant reduction in IL-1β and IL-6) by increasing the expression of tight-junction proteins that could maintain intestinal integrity [130]. Baicalin could also lower blood pressure by lowering calcium levels and causing vascular smooth muscle diastole [131]. In summary, baicalin has a combined effect on the prevention and treatment of cardiovascular disease.

## 6. Prospects

Baicalin, as one of the main active components of SBG, has attracted extensive attention in recent years due to its anti-inflammatory and lipid-lowering activities. Here, we review the anti-inflammatory effects of baicalin in the treatment of chronic inflammatory diseases such as steatohepatitis, RA, obesity, and type 2 diabetes; respiratory diseases; IBD; and cardiovascular diseases. Numerous pharmacological studies on baicalin have revealed a comparable molecular profile for effective disease treatment [65,79,90,107,132]. Specifically, baicalin was found to actively reduce both tissue and cellular damage by inhibiting the NF-κB pathway, downregulating inflammation-related factors’ expression (Figure 3), and reducing ROS production [17]. These significant actions ultimately help alleviate the inflammatory response and improve the oxidative stress state, thus offering essential therapeutic benefits. However, three further studies are still needed before baicalin could be applied to human disease treatment in a broad spectrum. The first should concern how to improve the bioavailability of baicalin more effectively; the second should concern how to objectively evaluate the chronic and acute toxicity of baicalin in different animal models; and the third should comprise more in-depth studies on the interaction of baicalin with other drugs, which are needed to mitigate or eliminate the adverse effects of drug interactions (Figure 4).

In recent decades, how to improve the bioavailability of baicalin has gradually been the focus of more researchers. At present, carrier delivery (nanocrystals and nano-emulsion) and structural modification (adding lipophilic groups) are used to improve the water solubility or fat solubility of baicalin. Although the new baicalin preparation improved oral bioavailability to a certain extent, the improvement had a limited effect on the fundamental solution of bioavailability due to the low absorption efficiency of baicalin. The limitation is that the new preparation only improves the solubility of baicalin but neglects the core key that the active compound penetrates the intestinal cell membrane, which is the ring-limiting step of drug absorption [21]. Incorporating lipophilic moieties into the spatial architecture of baicalin represents a promising strategy to enhance its lipid solubility and augment its capacity to traverse intestinal cell membranes. This approach seems to be efficacious and viable in optimal drug design. However, structural modification not only affected lipid solubility and drug stability but also affected other physicochemical properties, such as safety and biological activity [133]. In addition, there is a lack of research on whether structural modification can improve the bioavailability of baicalin in vivo. Objective evaluation of chronic and acute toxicity of baicalin in different animal models is also one of the main research contents. Although in vitro and in vivo tests have shown the safe dose range and action time of baicalin, more in-depth studies on its toxicity in different animal models should be conducted before its application in the treatment of human clinical diseases. Furthermore, it is imperative to pursue further development and enhancement of in vivo pharmacokinetic investigation of baicalin. These fundamental studies furnish scientific evidence to reinforce the mechanism of disease treatment.

## 7. Conclusions

In summary, baicalin is a multi-functional traditional Chinese medicine shown to have a mitigating effect on inflammation caused by liver, intestinal, and cardiovascular diseases in vitro and in vivo. Mechanism analysis of baicalin demonstrated that baicalin can regulate inflammatory responses induced by oxidative stress, and the regulation of oxidative stress mediated by baicalin is via PI3K/Akt/NRF2, kelch-like ECH-associated protein 1 (keap1), NF-κB, and HO-1 (heme oxygenase 1). Furthermore, the expression of IL-6, IL-1β, TNF-α, MIP-2, and MIP-1α is inextricably linked to the regulation of inflammation by baicalin. Considering the antibacterial and antiviral activity of baicalin, it is therefore a promising candidate for clinical inflammation treatment.

## Figures and Tables

**Figure 1 ijms-24-09317-f001:**
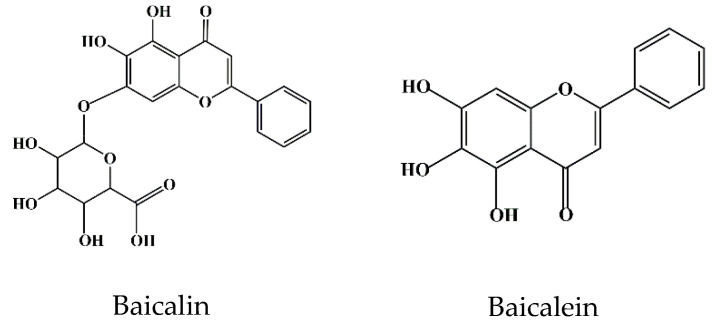
Chemical structure of baicalin and baicalein.

**Figure 2 ijms-24-09317-f002:**
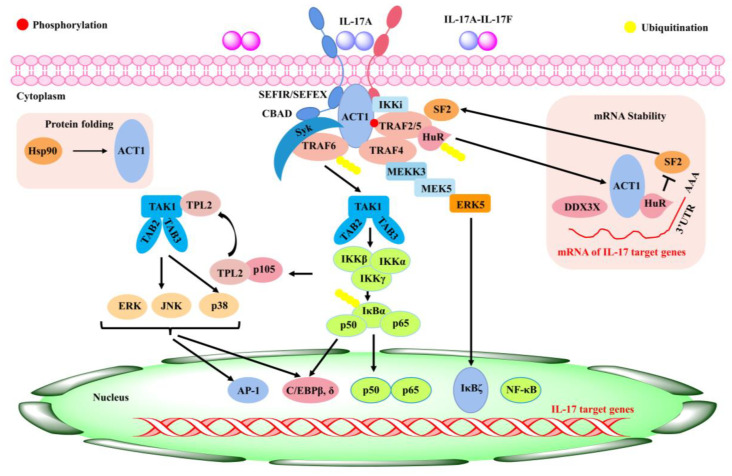
Inflammatory signaling mediated by IL-17 [78].

**Figure 3 ijms-24-09317-f003:**
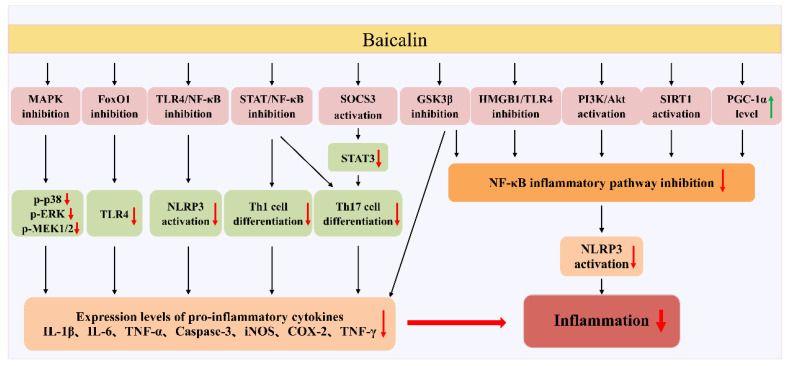
Schematic representation of anti-inflammatory pathways induced by baicalin. (1) The anti-inflammatory effect of baicalin depends on reducing the expression of inflammatory factors that might be involved in the regulation of several signaling pathways, including MAPK, FoxO1, TLR4/NF-κB, signal transducer and activator of transcription (STAT)/NF-κB, and suppressor of cytokine signaling 3 (SOCS3). (2) Baicalein inhibits NF-κB activation by activating phosphoinositide 3-kinase (PI3K)/protein kinase B (Akt) and sirtuin 1 (SIRTI) pathways and inhibiting the high-mobility group box 1 (HMGB1)/TLR4 signaling pathway. (3) Baicalin alleviates inflammatory responses by inhibiting NLRP3 inflammatory vesicle activation through negative regulation of the NF-κB pathway and glycogen synthase kinase-3 beta (GSK3 β) factor pathway.

**Figure 4 ijms-24-09317-f004:**
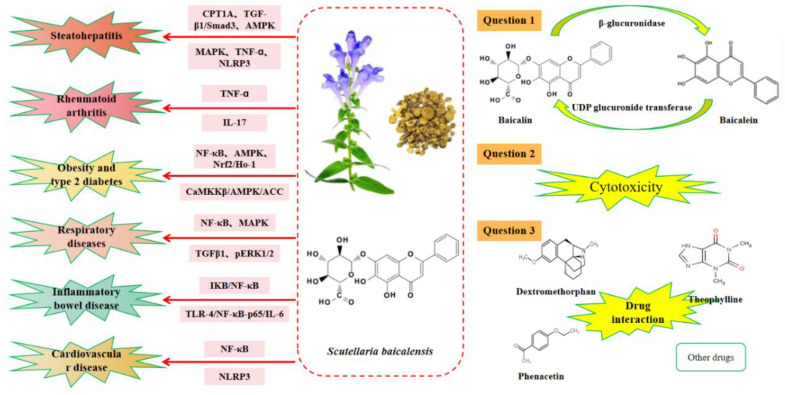
The anti-inflammation effects of baicalin and its unsolved question.

## Data Availability

There is no more data available.

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
