# Peer review of "The Pharmacological Efficacy of Baicalin in Inflammatory Diseases"

_ijms, 2023, doi:10.3390/ijms24119317_

Round 1

Reviewer 1 Report

The manuscript titled “The pharmacological efficacy of baicalin in inflammatory diseases” provides a review on baicalin potential to manage inflammation.

In order to improve the manuscript, the authors should address some points, namely:

1. Please revise English language and grammar, and minor typos (e.g.characteristics of growing season and cycle”; “natural flavonoids”; “The safe dose range of baicalin was suitable 100 for normal cell or body metabolism in different diseases” ).

2. Please revise if the genus is “Scutellaria” or “Scutellarin”;

3. Please revise sentences such as “harvested in autumn and grown for three years had the best quality”;

4. Please consider abbreviating the species name to “SBG”, for example, to improve the text making it easier to read;

5. Line 35, please revise the sentence “Flavonoid, as one of the most abundant natural compounds”, as flavonoids are a class of compounds;

6. Line 39, flavonoids are obtained from natural products, and thus are natural, there is no need to emphasize. Also, if they are natural, they are not developed. The authors may revise to are used to develop pharmaceutical formulations, or are used to treat some diseases.

7. Please revise this nomenclature “5,6, 7-trihydroxy flavone-7-beta-d-Glucuronic acid”, as there are more common notations;

8. Line 51/52 has repetitive content;

9. Figure 1 quality can be improved. Also, chemical structure representation should follow the same style in both structures.

10. Line 63, bioavailability can be assessed for phytochemicals in an extract for example, not necessarily in a drug formulation;

11. Lines 69-73, please revise if the sentences mentioning the two compounds are not contradictory;

12.  Figure 2 resolution should be improved;

13. The manuscript could benefit from the addition of a new figure elucidating the overall signalling of the inflammatory pathways, including the various pathways (with lesser detail), and the various cytokines, as well as an introduction to these pathways. This can help the reader to understand the studies referenced throughout the manuscript. Only IL-17 pathway is illustrated;

14. In addition to various reviews on baicalin bioactivities, other recent reviews focus on the same topic as this review, the anti-inflammatory activity (e.g. 10.1016/j.ejmech.2017.03.004 and 10.1016/j.jep.2021.114749). I believe the authors can improve their manuscript, by highlighting topics not discussed in previous publications, such as for example strategies to improve bioavailability, and targeted delivery, for example.

Overall, the manuscript presented is very interesting with significant a significant number of reviewed studies. It can benefit from revising the English language and various topics described above, as well as complementing the discussion.

English language should be revised to correct minor typos and grammatical consistency of some sentences. 

Reviewer 2 Report

Line 92: Typo in Cmax

Line no.38 

Line no.58: Add 'reducing/ inhibiting' inflammatory reaction.

Line 159: Better to have 'therapeutic' effect

Line 165: 'Excessing' should be replaced by excessive

Line 251 to 253: Rewrite

Author Response

Please see the attachment, thanks very much!

Round 2

Reviewer 1 Report

The authors have addressed the reviewers’ suggestions, answered all questions and improved their manuscript.

In my opinion, the manuscript can benefit from a second minor revision of the content soundness, to avoid sentences such as in line 37 “one of the most abundant flavonoids”. It should be clear that it is one of the most abundant flavonoids in the natural product discussed before, but not in all plants.

Also, as discussed in the first review, I believe the manuscript can be improved with additional information and analysis of current literature, distinguishing this manuscript from past reviews. I understand the point explained by the authors regarding this point, I believe the point mentioned in the authors' response could be added to the manuscript in my opinion. I will leave it to the Editor’s decision. 

Minor English Grammar and typos revision

Author Response

Please see the attachment, thank you!
